# Cancer-Associated B Cells in Sarcoma

**DOI:** 10.3390/cancers15030622

**Published:** 2023-01-19

**Authors:** Joseph K. Kendal, Michael S. Shehata, Serena Y. Lofftus, Joseph G. Crompton

**Affiliations:** 1Department of Orthopaedic Surgery, University of California, Los Angeles, CA 90404, USA; 2Jonsson Comprehensive Cancer Center, University of California, Los Angeles, Los Angeles, CA 90024, USA; 3Division of Surgical Oncology, Department of Surgery, University of California, Los Angeles, CA 90095, USA

**Keywords:** B cells, tertiary lymphoid structures, sarcoma

## Abstract

**Simple Summary:**

B cells are increasingly appreciated as important contributors to the tumor microenvironment in a myriad of cancer histologies, including sarcoma. In sarcoma, recent investigations have revealed associations between B cell expression signatures, the presence of tertiary lymphoid structures, and responses to immunotherapy. In this paper, we aim to provide a comprehensive review of the multiple putative roles of B cells in sarcoma, including a historical overview, an assessment of B cells within the sarcoma microenvironment, the role of tertiary lymphoid structures, the relationship between immunotherapy efficacy and B cell signatures, sarcoma antigens and anti-tumor antibodies, pro-tumor B cell relationships, and future research directions.

**Abstract:**

Despite being one of the first types of cancers studied that hinted at a major role of the immune system in pro- and anti-tumor biology, little is known about the immune microenvironment in sarcoma. Few types of sarcoma have shown major responses to immunotherapy, and its rarity and heterogeneity makes it challenging to study. With limited systemic treatment options, further understanding of the underlying mechanisms in sarcoma immunity may prove crucial in advancing sarcoma care. While great strides have been made in the field of immunotherapy over the last few decades, most of these efforts have focused on harnessing the T cell response, with little attention on the role B cells may play in the tumor microenvironment. A growing body of evidence suggests that B cells have both pro- and anti-tumoral effects in a large variety of cancers, and in the age of bioinformatics and multi-omic analysis, the complexity of the humoral response is just being appreciated. This review explores what is currently known about the role of B cells in sarcoma, including understanding the various B cell populations associated with sarcoma, the organization of intra-tumoral B cells in tertiary lymphoid structures, recent trials in immunotherapy in sarcoma, intra-tumoral immunoglobulin, the pro-tumor effects of B cells, and exciting future areas for research.

## 1. A Historical Perspective


*“It is by no means inconceivable that small accumulations of tumour cells may develop and, because of their possession of new antigenic potentialities, provoke an effective immunological reaction with regression of the tumour and no clinical hint of its existence. It has also been suggested that the result of surgery for cancer may to a large extent be determined by the degree of resistance, presumably immunological in nature, against the tumour cells.”*
*- Sir Macfarlane Burnet, 1957* [1]

In a sophisticated yet simple series of experiments in chickens, Max D. Cooper and colleagues reported in 1965 on a proposed “two-cell” system of antibody production, namely B and T cells [2,3]. These seminal experiments demonstrated that “B” cells derived from the avian bursa of Fabricius constitute the arm of the immune system that can produce antibodies and form necessary components of splenic follicles. As studies into antibody function followed, their presence was beginning to be recognized in the serum of patients with cancer. In one of the first studies on cancer-associated B cells (CABs) Donald Morton and Richard Malmgren reported in 1968 on a reproducible antibody response against osteosarcoma cells [4]. This antibody response was thought to be induced by a viral infectious agent due to the presence of antibody cross-reactivity among immediate family members and close associates. The authors presciently postulate that anti-tumor antibodies may be present in the general population; a hypothesis consistent with the more modern concept of cancer immunoediting [5]. Further investigation by Morton and Eilber at the National Institute of Health revealed the presence of anti-sarcoma antibodies derived from sarcoma patients [6]. These anti-sarcoma antibodies were found to bind to an antigen from a human liposarcoma cell line and were effective in ultimately inducing human complement fixation. Additionally, a study by McBride et al. in 1978 assessing early avian bursectomy revealed that a bursal-derived population of cells is essential in mediating resistance to Rous sarcoma virus-induced tumor formation, indicating an active role of B cells in anti-sarcoma immunity [7]. 

Early studies correlating humoral immunity dysfunction and cancer risk have raised questions about the role of B cells, anti-tumor antibodies, and complement fixation in mounting a response against sarcoma. As our understanding of B cell function evolves, it is becoming clearer that the human B cell compartment represents a complex, heterogenous group of cells with distinct phenotypic and functional subsets [8]. The role of B cells in cancer, and specifically in sarcoma, is increasingly appreciated, but the field of B cell immuno-oncology is still in its infancy.

## 2. B Cells in the Sarcoma Microenvironment 

### 2.1. Intra-Tumoral B Cells in Sarcoma

Like many intra-tumoral immune cells, B cells appear to have a variety of roles in tumor immunity (Figure 1, Table 1). While B cells have been studied in the context of tumor infiltration, comprehensive phenotypic and functional characterization is under active investigation [9]. In a systematic review, Wouters and Nelson evaluated the prognostic impact of CABs in multiple different primary cancer histologies, including soft tissue sarcoma (STS) [10]. On immunohistochemistry (IHC) assessment of CD20^+^ tumor infiltrating lymphocytes (TILs), the majority of cases (90.7%) demonstrated a positive or neutral impact of CD20^+^ TIL presence on patient prognosis. Tsagozis et al. assessed expression of the canonical B cell markers CD19 and CD20 in 33 patients with STS using IHC of whole tissue sections, and gene expression analysis using The Cancer Genome Atlas (TCGA) SARC cohort (265 patients total) for validation [11]. Multiplex IHC demonstrated a significantly positive prognostic impact of high CD20 expression, but CD19 was notably expressed at very low levels. The TCGA SARC data validation further demonstrated a positive prognostic effect of high and moderate CD20 expression; an effect that was attenuated in tumors expressing high levels of the immunosuppressive cytokine interleukin-10. CD20 expression was also correlated with both CD27 expression and CXCR3 expression, a correlation with which may indicate a role for intra-tumoral CD20^+^ cells in homing to inflamed tissue and T cell activation [11]. In a study assessing the tumor microenvironment (TME) of 54 conventional chordoma tissue samples, CD20 expression was graded moderate or prominent in 35% of patients [12]. Analysis of differentially expressed genes in conventional chordoma single cell analysis demonstrated gene ontology functional classifications in B cell activation and humoral immune response [13]. While some these findings have been corroborated by tissue microarray (TMA) analysis [14], Smolle et al. published conflicting data in 2021. A TMA on 1266 tissue core samples from 188 patients revealed that CD20^+^ cells had a negative correlation with local recurrence, and no influence on overall survival [15]. Discrepancies in results may be partly due to methodological differences, as Tsagozis et al. used whole tissue sections (compared to tissue cores), which may have increased the sensitivity of the assessment. These differences may also be due to the heterogeneity of CD20^+^ cells, and potential sampling artifacts. As knowledge of functional B cell sub-types improves, it has become clear that even amongst memory B cell populations there are as many as 10 functionally and phenotypically distinct populations [8]. Spatial distribution of intra-tumoral B cells also appears to be variable in sarcomas as well, with some tumor sub-types generally demonstrating a more diffuse lymphocytic infiltrate (for example, undifferentiated pleomorphic sarcoma, UPS) with low B cell presence, and others (for example, Rhabdomyosarcoma, RMS) demonstrating a more organized B cell presence in configurations known as tertiary lymphoid structures (TLSs) [16]. Mature TLSs can, however, also be found in UPS (Petitprez et al., extended data Figure 8) [17]. An analysis of TLS (discussed in detail below) indicates that the structure and spatial architecture of a B cell-mediated intra-tumoral immune response is more critical than the mere presence of these immune cell populations [18].

### 2.2. Tertiary Lymphoid Structures

Tertiary lymphoid structures (TLSs) are ectopic lymphoid aggregates (Figure 2) in a fibroblast network that form in non-lymphoid tissues and are induced by chronic inflammation and cancer (see reference 19 for recent review) [33,34,35]. While strict definitions of TLSs vary in the literature as these lymphoid aggregates may exist in a range of structural organizations, generally they are identified histologically by the presence of an inner B cell zone (cells expressing CD20) adjacent to a peripheral CD3^+^ T cell aggregate zone [17,34,35,36]. The cellular composition and structure of TLS greatly resembles that of a secondary lymphoid organ (SLO) and includes B cells that may or may not form germinal centers, follicular helper CD4^+^ T cells, populations of CD4^+^ helper T cells, CD8^+^ cytotoxic T cells, follicular dendritic cells that promote germinal center formation and memory B cell selection, mature dendritic cells that present antigen to T cells, a dense stromal network of follicular reticular cells that provides the structure for this cellular organization, and high endothelial vessels that are key in the recruitment of immune cells to the TLS. There are some differences in these populations across different cancer types as well as differences in the degree of organization, which may suggest differences in the immunogenicity of certain tumors. Even the most well-organized TLSs with active germinal centers lack clearly defined light zones and dark zones, which are critical to affinity maturation in SLOs [35]. This may suggest differences in the repertoire of antigen specificity of B cells that have undergone selection in TLS [35]. Another major difference in the TLS structure when compared to SLOs is the absence of a capsule, which may result in increased antigen sampling of the TME by APCs, as well as the absence of subcapsular macrophages [37]. 

Despite these differences, TLSs have been shown to be functionally similar to SLOs, with evidence for clonal proliferation, isotype switching, somatic hypermutation, B cell effector differentiation, and T cell activation, all of which may have significant implications for anti-tumor immunity [35]. Accordingly, there is an increasing appreciation for the anti-tumor potential of TLSs, including a strong role for B cells; specifically, antibody and pro-inflammatory cytokine-producing plasma cells. For instance, in ovarian cancer, presence of plasma cells and TLSs was correlated with increased cytotoxic T cell activity and increased survival [38]. In breast cancer, the presence of CABs is positively correlated with co-presence of CD3^+^ TILs and the number of TLSs present [9]. These anti-tumor effects of TLSs are thought to be mediated by dendritic cell tumor antigen presentation to T cells, followed by the activation of both T and B cell effectors [39]. B cell activation in the context of a TLS ultimately can result in differentiation to a memory B cell phenotype and anti-tumor antibody producing plasma cells [40]. 

Further phenotypic and functional characterization of B cells within TLS in lung cancer revealed a wide spectrum of B cell maturation, including plasma cell presence, evidence of somatic hypermutation, class-switch recombination, and ultimately immunoglobulin production, confirming TLS contribution to anti-tumor humoral immunity [9,41,42,43,44,45]. In renal cell carcinoma, IgA- and IgG-positive plasma cells derived from the TLS appear to disseminate into the tumor through CXCL-12-expressing fibroblast tracts, with TLS^+^ tumors exhibiting high frequencies of IgG-stained and apoptotic malignant cells, suggestive of anti-tumor effector activity [40]. Taken together, perhaps not surprisingly, there has been a positive prognostic impact of TLSs repeatedly observed in multiple primary tumor types, including melanoma, lung, pancreatic, breast, renal cell, and colorectal cancer [9,41,42,43,44,45]. In sarcoma, recent efforts have revealed that TLS expression signatures in primary tumors is positively correlated with patient survival as well as responses to the immune checkpoint blockade (ICB, discussed below). The presence of mature TLSs was confirmed in UPS, de-differentiated LPS, and leiomyosarcoma (LMS) via IHC [17]. The presence of TLSs in metastatic deposits appears to be tumor dependent, with osteosarcoma and LMS lung metastases demonstrating the low density of lymphoid aggregates, as determined by CD20^+^ follicles and the number of mature dendritic cells [35]. 

Another putative function of CABs in TLSs is in the presentation of antigens to T cells [27,46]. B-T cell antigen presentation has been well described in SLOs (spleen and lymph nodes) [21] and may also prove to be a mechanism of anti-tumor action of CABs in TLS. Rubtsov et al. demonstrate that specific B cell subsets can act as potent antigen-presenting cells, priming CD-4 T cells [21]. Corroborating these results, other studies have demonstrated that activated B cells may act as MHC class-II-dependent antigen presenting cells to activate anti-tumor T cell responses [23,47]. In a study assessing the cancer-testis antigen New York esophageal squamous cell carcinoma 1 (NY-ESO 1, expressed in STS, particularly synovial sarcoma), B cells and other professional antigen-presenting cells rapidly internalize the antigen and effectively cross-present an immunogenic epitope via MHC class I [24]. Tumor-infiltrating B cells isolated from non-small cell lung cancer were shown to present antigen to CD-4 T cells more efficiently than dendritic cells in vitro [23]. Antigen capture and endocytosis have been shown to be efficient and rapid, and are dependent on binding to the B cell receptor [22]. Taken together, it is possible that B cells in the tumor microenvironment may function to help prime an anti-tumor T cell response by presenting tumor antigens (Figure 1).

Despite these promising findings suggesting a robust anti-tumor role for TLSs, the maturity of TLSs may have a significant effect on these effector responses [18]. While Cabrita et al. demonstrated a prognostic association between mature TLSs and survival in melanoma, TLSs that were poorly organized and less mature demonstrated dysfunctional T cell expression profiles [20]. The effects of these complex interactions between tumor cells and these collections of lymphoid cells are nuanced, and better understanding of the processes that drive TLS maturation in the TME is warranted.

## 3. Sarcoma Immunotherapy and B Cells 

In a report from 1891, Dr. William Coley (a sarcoma surgeon and the “father of cancer immunotherapy”) documented cases of sarcoma tumor regression after erysipelas, and subsequently developed Coley’s toxin vaccine, which is a mixture of heat-inactivated Streptococcus pyogenes and Serratia marcescens, and one of the first attempts at cancer immunotherapy [48,49]. Despite early evidence of sarcoma tumor immunology, evasion of the immune system was not described as a hallmark of cancer until 2011 [50], and investigation into immunotherapies has since advanced at a rapid pace. One such advancement in immunotherapy has been the development of ICB, which has demonstrated substantial promise in multiple solid cancer sub-types. D’Angelo et al. assessed programmed death ligand 1 (PD-L1) expression in 50 heterogeneous sarcoma samples using IHC and identified PD-L1 expression in only 12% of cases, but tumor infiltrating lymphocytes were present in 98% of samples [51]. Data summarized from three STS immunotherapy clinical trials demonstrated PD-L1 expression in 13.6% of cases (21/154 patients) [52]. Despite these findings, use of monotherapy ICB (antibodies against PD-1, PD-L1 or CTLA-4) in sarcoma has traditionally shown poor efficacy, dependent on the sarcoma histologic subtype [53,54,55]. Sarcomas in general have a low tumor mutational burden (TMB) and are thought to be relatively immune inert (in comparison to histologies such as melanoma), which is an important TME characteristic that is associated with response to ICB [56,57]. The SARC-028 trial and subsequent expansion cohort shed more light on the effect of ICB in bone and soft-tissue sarcoma [58,59]. In this non-randomized phase two trial, patients received the programmed death one (PD-1) monoclonal antibody, Pembrolizumab. Clinical activity was relatively limited to those with undifferentiated pleomorphic sarcoma (UPS) and liposarcoma (LPS) with 9/40 (23%) and 4/39 (10%) patients, respectively, demonstrating a partial or meaningful response to treatment. Pooled data from nine multicenter trials assessing ICB (PD1/PD-L1 pathway) in sarcoma revealed an objective response rate of 15.1% and non-progression rate of 58.5% for all sub-types [52]. Further insights into ICB efficacy in sarcoma have been more recently revealed through investigation of sarcoma samples with TLS and B cell signatures, as discussed below. 

### Tertiary Lymphoid Structures and Immunotherapy

Much research investigating correlations between ICB response and tumor immune infiltrates has focused largely on the presence of CD8^+^ T cells as a marker of response [60]. Indeed, in a tissue assessment of SARC-028 trial patients, Keung et al. reported that the presence of CD3^+^CD8^+^PD-1^+^ T cells and tumor-associated macrophages (TAMs) expressing PD-L1 corresponded with response to ICB and improved progression-free survival [61]. B cell presence was not assessed in this study; however, recent global analyses of the sarcoma tumor microenvironment have delineated some very intriguing relationships between sarcoma response to ICB and the presence of intra-tumoral B cells. The PD-1 pathway has been traditionally assessed in T cell regulation; however, B cells can express PD-1, and T-follicular helper cells, which promote B cell immune response, also express PD-1 [62]. Petitprez et al. performed an in-depth assessment of the TME in STS, and using computational methods from bulk RNA sequencing data they identified the presence of five transcriptionally distinct immune signatures, which they designated as separate sarcoma immune classes (SICs) [17]. A particular SIC with a strong B cell lineage expression profile was found to be significantly associated with improved patient survival, as well as improved response to ICB (based on tissue analysis from the SARC-028 dataset). B cell expression was the strongest predictor of improved survival, and this relationship was maintained independent of cytotoxic T lymphocyte abundance or other immune signatures. The association between over-representation of intra-tumoral B cell gene expression signatures and response to immune checkpoint blockade (ICB) has also been observed in other types of cancers [63]. For example, clonal expansion and enrichment of functional B cells is evident in patients with melanoma and renal cell carcinoma (RCC) who respond to ICB [63,64,65]. While Helmink et al. identified unique intra-tumoral memory B cell populations specific to ICB responders in melanoma samples using mass cytometry (for example, CXCR3^+^ switched memory B cells), a high-dimensional analysis of B cells in sarcoma samples has not yet been reported [63]. 

The study by Petitprez et al. Also offers clues that suggest intra-tumoral B cells are key participants in ICB response through TLS formation. In the class of tumors that demonstrated a robust B cell expression signature and response to ICB, there was specifically an elevated expression of a plasma cell signature as well as CXCL13 expression, both of which are increasingly appreciated as facets of TLS [32,66]. Immunohistochemistry identified TLSs in 82% of tumors from this SIC, where only one sample had TLSs from all four of the other SICs combined [42]. The presence of TLSs and elevated TLS expression signatures correlating with ICB response has also been appreciated in other tumor types, such as renal cell carcinoma, melanoma, and bladder cancer (reviewed in Kinker et al.) [64].

Based on the observation that sarcoma samples with a rich B cell transcriptomic signature and TLS presence may represent a population of patients with more favorable responses to ICB, Italiano et al. investigated whether the presence of TLSs is a potential biomarker for ICB response [32]. The PEMBROSARC trial was a phase II clinical trial assessing the efficacy of ICB in combination with cyclophosphamide in STS and demonstrated overall limited efficacy [67]. When selecting patients with TLSs in an amended cohort, however, the six-month non-progression rate was a relatively impressive 40% (vs. 4.9% in the original cohort), and the objective response rate was 30% (vs. 2.4% in the original cohort), demonstrating the dramatic predictive value of TLS presence on response to ICB in this STS cohort [32]. In this amended trial, TLSs were defined histologically as areas of intra-tumoral CD3^+^/CD20^+^ aggregates with at least 700 cells, and were identified on initial screening in 20% of STS cases. 

## 4. Sarcoma Antigens and Anti-Tumor Antibodies 

The presence of antibodies in tumor-associated antigens, or new epitopes from alternatively processed proteins has been demonstrated in several different types of malignancies, including breast cancer, hepatocellular carcinoma, and lung cancer [68,69,70,71]. The inconsistent presence of a humoral response in cancer may be due to the diversity of autoantigens, intrinsic tumor features, weak tumor antigen expression patterns, and MHC variability across individuals [72]. Moreover, in many cases it is unclear whether the presence of auto-antibodies corresponds to an active immune surveillance phenomenon, an anti-tumor effect, or loss of self-tolerance in a cancer host [73]. Auto-antibodies may indeed have many beneficial anti-tumor effects, including the induction of antibody-dependent, cell-mediated cytotoxicity (particularly IgG1 mediated), complement-mediated cytotoxicity, and T cell activation through antigen cross-presentation [27,72]. A recent analysis of ovarian carcinoma revealed that anti-tumor auto-antibody production has a positive prognostic correlation, and these self-reactive clones can arise from both antigen-driven selection through somatic hypermutation and from tumor-binding, germline-encoded immunoglobulin [74].

### 4.1. Cancer-Testis Antigens and Translocation-Positive Sarcoma

Cancer-testis antigens (CTA) have been investigated as potential immunogenic targets in sarcoma. CTAs are proteins expressed in the placenta, testis, and embryos, and are then silenced in non-immune privileged healthy tissues and ultimately, re-expressed in certain malignancies, including bone and soft-tissue sarcomas. Examples of CTAs expressed in sarcomas include NY-ESO-1, melanoma-associated antigen (MAGE), preferentially expressed antigen of melanoma (PRAME), and synovial sarcoma X (SSX) [75]. Expression of these CTAs is variable depending on the sarcoma subtype and particular CTA. Various sub-types of MAGE are highly expressed in osteosarcoma, synovial sarcoma, and myxoid liposarcoma [76,77,78,79]. Tumor tissue specificity is a critical characteristic to assess when analyzing CTAs such as MAGE, as collateral damage of off-target effects of targeted treatment such as neurologic toxicity may be catastrophic [80]. NY-ESO-1 is a CTA of particular interest as it has notable expression in synovial sarcoma and myxoid liposarcoma [77,81]. When assessing NY-ESO-1 expression in myxoid liposarcoma using a comprehensive tissue microarray panel, Shurell et al. revealed expression in 100% of tested tumors [81]. Despite CTAs offering the enticing potential for cancer-specific antigenicity, these proteins may not be intrinsically very immunogenic. In a study assessing the cancer immunome of 54 sarcoma patients, an antibody response to CTAs was only present in 2 patients [82]. The response in these two patients was restricted to the NY-ESO-1 CTA, which is expressed by almost all these tumors, suggesting only a small portion of these differentially expressed proteins can generate an immune response. This is intriguing as it is unclear if this points towards an anti-immune mechanism by the tumor cells that promotes tolerance, or if CTAs are well tolerated given their natural expression in germ tissues. In this study, there was infrequent overlap of immunomes between sarcoma patients, which is reflective of the inherent heterogeneity of sarcoma sub-types and demonstrates that using CTAs as a universal target for sarcoma may be challenging. Given the restricted tumor specificity of CTA expression and potential therapeutic targeting, attempts at tumor vaccination targeting CTAs in sarcoma are ongoing. For example, one such trial proceeded with tumor vaccination using a lentiviral vector, targeting dendritic cells to present the antigen NY-ESO-1. This strategy induced an anti-NY-ESO-1 antibody response in only 4% of sarcoma patients; however, overall survival was significantly higher in patients with both pre-existing anti-sarcoma antibodies (39% of sarcoma patients with NY-ESO-1^+^ tumors), or those with induced antibody production after vaccination [83]. This raises questions as to optimizing strategies to induce a more robust anti-CTA antibody response, which may prove to have a positive influence on sarcoma progression. 

Another potential immunogenic sarcoma antigen is fibroblast activation protein (FAP) [84,85]. FAP is a membrane serine protease physiologically expressed in the granulation tissue of healing wounds, but is also expressed in both cancer-associated fibroblasts and sarcoma cells. Knockdown of FAP expression in osteosarcoma cells reduces proliferation and migration [85]. FAP has been a target of interest for anti-tumor vaccination in a strategy to target both tumor cells and immunosuppressive cancer-associated fibroblasts [86]. Another category of potential sarcoma-specific antibody targets are the neoantigens that occur from the fusion proteins expressed secondary to canonical translocations in sarcomas. These make attractive vaccination targets as they are truly neoantigens that are otherwise not expressed in normal tissue; however, their immunogenicity and efficacy may be variable. SYT-SSX represents a sarcoma fusion protein that is expressed in synovial sarcoma secondary to the translocation of one of several highly homologous CTAs on chromosome X (SSX1, SSX2 and SSX4), and the SYT gene on chromosome 18, t(X;18) [87]. A clinical trial assessing the safety and efficacy of a SYT-SSX peptide vaccine for patients with metastatic synovial sarcoma demonstrated safety and variable tumor responses [88]. A similar approach in alveolar RMS (PAX3/FKHR translocation positive) in conjunction with interleukin-2 infusion demonstrated poor efficacy [89]. The EWS-FLI1 (t;11:22) translocation is present in over 85% of the Ewing sarcoma family of tumors and represents a tumor specific protein that has been previously targeted using sequence-specific small interfering RNA [90]. Liu et al. took a somewhat divergent approach to identify two immunogenic B cell epitopes of the EWS-FLI1 fusion protein, which could subsequently be used for the development of a B cell-activating, dendritic cell vaccine [91]. The authors postulate that using peptides that act as both T and B cell epitopes when developing DC vaccines can optimize a more robust immune response, and in theory, anti-tumor antibody production while minimizing any side effects that may occur when using an entire fusion protein to generate a vaccine. Cancer vaccine research has traditionally focused on dendritic cell-mediated T cell activation, with less emphasis on identifying B cell epitope cancer vaccines [92]. Of note, T cell exhaustion can occur and can limit the effectiveness of CTA T cell vaccines [93]. B cell vaccination success relies on high levels of surface antigen expression by tumor cells, as well as antibody specificity to the antigen to mediate ADCC [94]. Efforts at B cell immunization attempts therefore must be aimed at inducing robust antibody production against a highly expressed surface peptide. Passive immunity with humanized monoclonal antibodies has demonstrated substantial success in the clinical setting in certain cancers, including the development of Trastuzumab against the HER-2 breast cancer mutation, and rituximab (humanized anti-CD20 antibody) in lymphoma, as ADCC and other complement-mediated cytotoxicities have been shown to be critical to their efficacy in addition to direct cell signaling effects from antibody binding [94]. Monoclonal antibodies, however, are costly and have a limited duration of action due to the half-life of IgG, necessitating repeated dosage [92]. The use of vaccines encompassing a conformational B cell epitope with a promiscuous T cell epitope is a proposed approach to induce anti-tumor B cell- and T cell-mediated immunity [92,95]. Given the prevalence of either CTA or translocation-positive sarcoma antigens, further attempts at optimizing anti-tumor B cell-mediated vaccinations are warranted.

### 4.2. Intratumoral Sarcoma Immunoglobulin 

Intra-tumoral immunoglobulin (Ig) have long been identified in the tumor microenvironment as evidence of a tumor-induced humoral response [96]. Presence of intra-tumoral Ig has been correlated with survival in melanoma and non-small cell lung cancer, corroborating findings that plasma cell infiltration has positive prognostic associations [97,98,99]. Intra-tumoral Ig demonstrate tumor specificity and limited clonality with evidence of somatic hypermutation, suggesting a localized selection and expansion process, and further evidence for the recapitulation of SLO functions in TLSs [100,101].

The functionality of antibodies is largely determined by their isotype class, and several studies have identified a prognostic correlation with various immunoglobulin isotypes in other cancers, adding further complexity to understanding the role of intra-tumoral B cells. IgG1 is a strong inducer of Fc-mediated effector mechanisms, including antibody-dependent cellular cytotoxicity, complement-dependent cytotoxicity, and antibody-dependent cellular phagocytosis [25]. Isaeva et al. determined that in lung adenocarcinoma, high intra-tumoral production of IgG1was correlated with improved prognosis in KRAS driver mutation-positive tumors [102]. Similar findings were observed in melanoma, where high intra-tumoral expression of immunoglobulin heavy chain (IgH), high IgH clonality, and a high proportion of IgG1 of all IgH transcripts were all associated with improved prognosis [99]. These data support an anti-tumor effector function of intra-tumoral IgG1. Evidence for anti-tumor effects of IgG1 in sarcoma can be seen in the amended PEMRBOSARC trial, whereby expression of IgG and plasma cell enrichment was seen in responders to pembrolizumab and cyclophosphamide [32]. A recent study analyzing high-grade ovarian cancer identified a positive prognostic significance in the presence of intra-tumoral IgG-producing cells and the coating of tumor cells with IgG [74]. However, when they further analyzed different solid cancer types for IgG coating of tumor cells, liposarcoma was found to have a high IgG coating and fibrosarcoma had very little, which they termed “antibody non-reactive.” It is possible that grade or histology have an impact on the reactivity to IgG in various sarcomas.

IgG4, which does not have strong Fc binding, has been hypothesized to act as a blocking antibody and impair IgG1 effector functions [26]. While intra-tumoral IgG4 has demonstrated an association with increased IL-4 and IL-10 production and decreased ADCC and antibody-dependent cellular phagocytosis of tumor cells in melanoma, higher levels of IgG4 had positive prognostic associations in lung squamous cell carcinoma and sub-types of lung adenocarcinoma [102,103,104]. IgA may function similarly to IgG1 via Fc-mediated effector mechanisms; however, its expression is promoted by TGF-β, and IgA^+^ B cells have been found to upregulate regulatory T cells. IgE, which is promoted by IL-4, drives CD4^+^ T cells to Th2 fate, which has been associated with decreased cytotoxic T cell activity. IgD can activate basophils to produce more IL-4 and has been associated with increased class switching towards IgE and IgA. Taken together, IgA, IgE, and IgD may be markers of immunosuppression and pro-tumor immunity [27]. Indeed, IgE and IgD have been identified as negative prognostic markers in melanoma [99]. IgA presence or high IgA/IgH proportions were associated with worsened patient prognosis in lung cancer and bladder cancer [102,105]. Although no studies to date have identified IgA, IgE, or IgD in the microenvironment of sarcoma, increased T_reg_ infiltration has been associated with increased rates of local recurrence after resection [106]. Exploring the relationship between IgA-expressing B cells and T_regs_ in the sarcoma microenvironment may offer clues into the pro-tumor effects of B cells. 

While the majority of intra-tumoral immunoglobulin is presumably expressed by CABs, there has been increasing appreciation that immunoglobulin may be expressed by cells other than B cells. A few groups have identified immunoglobulin in the expression profiles of various carcinomas where it has been hypothesized to act as a growth factor [107]. Tumor antibody production has also been identified in various sarcoma sub-types, with elevated levels of IgG expression in sarcomas compared to benign tumor histologies, and a correlation with IgG staining with markers of tumor proliferation [108]. Increasing immunoglobulin kappa chain expression was seen in higher grade sarcomas (grade 3), vs. grade one and two sarcomas, raising questions about a relationship between tumor cell Ig production and prognosis. The intrinsic ability to generate IgG in sarcoma appears to be similar to the underlying mechanisms of IgG production in B cells, including shared expression of essential enzymes for somatic hypermutation and Ig class switching (recombination activating genes one and two, and activation-induced cytidine deaminase), as well as cell membrane localization of IgG [109]. It is unclear if malignant, non-lymphocyte Ig production is an adaptive mechanism by tumor cells, or a feature of greater dysplasia and irregular expression. How tumor-expressed Ig affects sarcoma tumor biology and response to therapy is still unclear, but may it confound findings when attempting to study the pro-tumor or anti-tumor effects of Ig produced by intra-tumoral B cells. The distinction between Ig production from sarcoma cells and tumor infiltrating B cells, as well as Ig isotyping and spatial intra-tumoral relationships, will be critical in determining precise roles of intra-tumoral Ig in sarcoma biology.

## 5. Pro-Tumor Relationships of B Cells in Sarcoma

Some studies have identified a more nefarious relationship of B cells in the cancer setting (including in sarcoma). Under certain conditions, such as immature development of TLSs or recruitment of regulatory B cells, B cells may in fact promote tumor progression [18,27,29]. Regulatory B cells (B_reg_) have been found to be the source of the immunosuppressive cytokine IL-10 [30]. B_reg_ cells can dampen effective anti-tumor responses by several mechanisms, including immunosuppressive cytokines (IL-10, IL-35 and TGF-b), activation of immune checkpoint pathways, and secretion of ineffective antibodies [27,28]. Maglioco et al. determined in a pre-clinical methylcholanthrene-induced murine fibrosarcoma model that anti-CD20 antibody-mediated B cell depletion after the tumor was established decreased tumor growth [29]. The mechanism of B cell-mediated tumor promotion in this model was linked to an increase in T_reg_ number and inhibition of effector T cell function [29,31]. In contrast, when B cells were depleted prior to sarcoma establishment, an increase in tumor growth was appreciated, suggesting divergent effects on B cell presence during tumor establishment and subsequent growth. Similar results have been reported in various pre-clinical cancer models [28]. In another set of experiments assessing immune re-polarization via B_reg_ targeting, Premkumar and Shankar evaluated TGF-b signaling inhibition in a murine fibrosarcoma model [110]. In this study, the immunosuppression induced by the regulatory B-T cell axis was reduced by TGF-b inhibition and ultimately led to a decreased fibrosarcoma tumor burden. With regards to pro-tumor relationships of B cells in human sarcoma tissue, as discussed earlier, Smolle et al. identified a negative correlation between the presence of intra-tumoral CD20^+^ cells with local recurrence and no influence on overall survival [15]. 

As discussed by Fridman et al., an important consideration when synthesizing information from murine models of cancer is the method of tumor establishment [111]. Cell line-derived syngeneic models have several advantages in studying immuno-oncology in a pre-clinical setting; however, the use of a homogeneous cell line tumor graft with rapid tumor establishment may not be sufficiently immunogenic to promote formation of TLS [111]. A potential solution is to study induced spontaneous tumor development. For example, TLS development has been identified in the Kras/p53 (KP) model of virally delivered Cre-recombinase-mediated authochonous in lung cancer [112]. In a KP model of pancreatic ductal adenocarcinoma, Delvecchio et al. were able to induce TLS formation using lymphoid chemokine intra-tumoral injection (using CXCL13) [113]. In this syngeneic model, TLS induction potentiated the action of gemcitabine chemotherapy. Similarly, the development of a KP STS model has facilitated the study of sarcoma immunology in a syngeneic setting [114,115]. Use of such models will facilitate the assessment of conventional sarcoma therapies, such as radiation therapy, in conjunction with immunopotentiating treatments to study TLS formation and effects in sarcoma. 

## 6. Next Steps

As the field of B cell immuno-oncology matures, there are several lines of active investigation to improve knowledge of intra-tumoral B cells and potential anti-sarcoma therapeutic approaches. As discussed above, future pre-clinical investigations aimed at the induction of intra-tumoral lymphoid neogenesis through combination therapies incorporating current standard of care treatments, including radiation therapy (RT), may provide valuable insights. Such examples include interventional immunotherapy with Stimulator of Interferon genes (STING) agonists, which have been found to induce therapeutic formation of TLSs [116]. Another approach discussed above includes intra-tumoral injection of cytokines (such as CXCL-13) to stimulate TLS formation and synergize with systemic therapy [113]. Yagawa et al. assessed candidate cytokines to generate ectopic TLSs and identified CXCL-12 and CXCL-13 as chemokines that induce concentration-dependent and significant B cell chemotaxis [117]. Reprogramming immunologically “cold” sarcomas to a more favorable microenvironment, such as macrophage re-programming using anti-CD47 [118], may prove to be synergistic with TLS formation and effector function. An intriguing ex vivo approach to study rare cancers, such as sarcoma and chordoma including the development of tumor organoids, which accurately recapitulate parent tumor features [119]. Generation of ectopic lymphoid structures using a sarcoma organoid template remains a nascent field in B cell immuno-oncology. Further potential areas of investigation include the generation of next-generation anti-sarcoma vaccines incorporating B cell immunity, discussed above. B cell engineering also remains scarcely investigated in sarcoma. Li et al. inhibited pulmonary metastases in a murine model of breast cancer (41T cell line) using transfers of tumor-primed and in vitro-activated B cells. They found that the anti-metastatic effects were synergistic when B cells were co-transferred with activated T cells [120]. In this study, adoptive B and T cell transfer induced an effective anti-tumor immune response, which in B cells included activation of effector T cells, production of anti-tumor antibodies, and direct anti-tumor cytotoxic effects. Rossetti et al. demonstrated similar results in a model of cervical cancer [47]. Further studies in adoptive B cell transfer and ex vivo priming are warranted. Predicting response to therapy is another clinically relevant direction, as demonstrated in the PEMBROSARC TLS cohort, which may also prove to be a valuable pathway towards optimized personalized therapy [32].

## 7. Conclusions

In summary, sarcomas are rare, aggressive, heterogeneous mesenchymal malignancies that generally have high resistance to traditional therapies. Novel therapeutic approaches are needed, and recent advances in immuno-oncology have provided early scientific promise. Early studies investigating humoral immunity in the context of sarcoma raise questions about the potential of using B cell immunotherapeutic approaches in treating sarcoma. Continued efforts in understanding the branching oncologic aspects of Max Cooper’s original two-cell system of antibody production will yield further valuable knowledge to direct potentially actionable therapeutic advances in sarcoma immunology. 

## Figures and Tables

**Figure 1 cancers-15-00622-f001:**
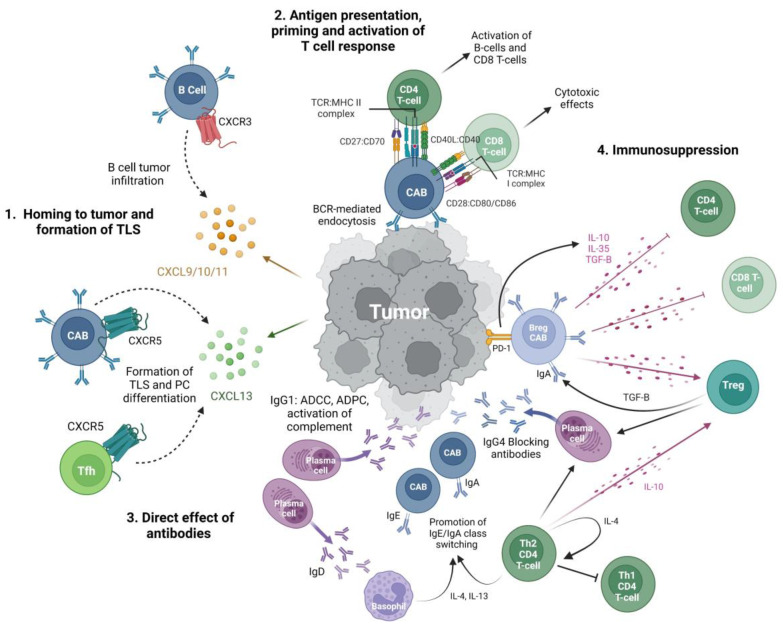
Putative effector functions of B cells in sarcoma. 1. Homing to tumor and formation of TLS: increased expression of CXCR3 in intra-tumoral B cells suggests homing to tumor via CXCR3:CXCL9/10/11 pathways [11,19]. Increased CXCL13 in the tumor microenvironment promotes the homing of B cells and follicular T helper cells via CXCR5 and formation of TLS [20]. 2. Antigen presentation, priming and activating T cell response: B cells act as professional APCs that use BCR-mediated endocytosis to process and present tumor antigens via MHC class I and II molecules to T cells [21,22]. Relatively high levels of expression of costimulatory molecules CD27, CD80, and CD86 suggest intra-tumoral B cells are potent T cell stimulators which have downstream anti-tumor effects [11,23,24]. 3. Direct effects of antibodies: differentiation of B cells into plasma cells generates antibodies with a variety of effector functions. IgG1 can opsonize tumor cells for antibody-dependent phagocytosis, antibody-dependent cellular cytotoxicity, and activation of the complement [25]. IgG4 has been hypothesized to act as a blocking antibody and inhibit IgG1 activity and is promoted by T_regs_ and Th2 CD4^+^ T cells [26]. IgA may act via Fc receptor-mediated effector functions such as IgG1, however, IgA-positive B cells also participate in a positive feedback loop with immunosuppressive T_regs_, as TGF-β secreted by T_regs_ promotes IgA class switching and IgA^+^ B_regs_ secrete IL-10 and IL-35 [27]. IgD promotes class switching towards IgA and IgE via basophil activation. High IgE reflects increased Th2 differentiation of CD4^+^ T cells and limited differentiation towards Th1 with decreased anti-tumor, cell-mediated responses [27]. 4. Immunosuppression: B_regs_ may promote tumor growth by secretion of immunosuppressive cytokines that downregulate T cell effector functions and promote T_reg_ differentiation and proliferation. B_regs_ are IgA^+^ and can express PD-1, which acts to promote IL-10 production [27,28]. This may have implications for use of immune checkpoint inhibition in sarcoma treatment [28,29,30,31]. Abbreviations: CAB: cancer associated B cell; B_reg_: regulatory B cell; T_reg_: regulatory T cell; APC: antigen presenting cell; Tfh: T follicular helper cell; PC: plasma cell; BCR: B cell receptor, TLS: tertiary lymphoid structure. *Created with BioRender.com*.

**Figure 2 cancers-15-00622-f002:**
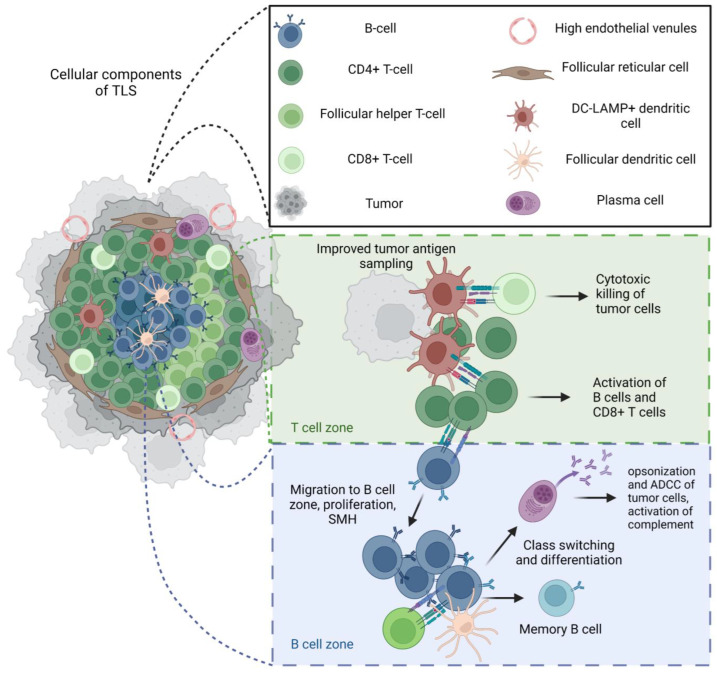
Structure and function of mature tertiary lymphoid structures (TLS). Intra-tumoral B cells in sarcoma may be scarce and diffuse, and/or associated with organized anti-tumor immune responses within tertiary lymphoid structures. When mature, TLSs have been associated with a positive prognosis and responses to checkpoint inhibitions in sarcoma [36]. The cellular composition and structure of a mature TLSs greatly resembles that of a secondary lymphoid organ (SLO) and includes B cells that may or may not form germinal centers, follicular helper CD4^+^T cells, a populations of CD4^+^ helper T cells, CD8^+^ cytotoxic T cells, follicular dendritic cells that promote germinal center formation and memory B cell selection, mature dendritic cells that present antigens to T cells, a dense stromal network of follicular reticular cells that provides the structure for this cellular organization, and high endothelial vessels that are key in the recruitment of immune cells to the TLS. Mature TLSs in tumors also recapitulate many functional mechanisms of SLOs, with evidence of T cell activation, somatic hypermutation (SMH), class switching, differentiation of B cells into plasma cells, and memory cells. Given the lack of capsule, it is possible that improved tumor antigen sampling can result in effective anti-tumor immunity through the activation and differentiation of B and T cells in the tumor microenvironment; however, the effectiveness of these mechanisms seems to be related to the maturity of the TLS, highlighting the importance of all these complex cellular interactions in a robust anti-tumor response [17,20]. *Created with BioRender.com*.

**Table 1 cancers-15-00622-t001:** Summary table of B cell presence in sarcoma clinical tissue. Table includes both retrospective statistical analyses using established cancer genomic databases as well prospective data. Abbreviations: GEO, gene expression omnibus; GIST, gastrointestinal stromal tumor; IHC, immunohistochemistry; RMS, rhabdomyosarcoma; SARC TCGA, the Cancer Genome Atlas, sarcoma; STS, soft tissue sarcoma; UPS, undifferentiated pleomorphic sarcoma.

Study	Histology	B Cell Marker	Detection Method	Number of Positive Samples	Prognostic Association
Sorbye [14]	Mixed non-GIST STS	CD20	IHC	22/105	Positive correlation with disease-specific survival
Tsagozis et al. [11]	Mixed STS	CD20	IHC	13/30	Positive correlation with metastasis-free survival and overall survival
Tsagozis et al. [11]	Mixed STS (SARC TCGA dataset)	MS4A1 RNA expression	Transcriptional data from SARC TCGA	Not reported	Overall survival
Zou et al. [12]	Chordoma	CD20	IHC	19/54	None
Smolle et al. [15]	Mixed STS	CD20	IHC	*n* = 188, number positive not reported	Negatively associated with local recurrence
Chen et al. [16]	Rhabdomyosarcoma and UPS	CD20 and CD3 to identify TLS	IHC	UPS: 2/34 RMS: 19/47	Not assessed
Petitprez et al. [17]	Mixed STS (SARC TCGA, GEO accessions GSE21050, GSE21122 and GSE30929 datasets)	BANK1, CD19, CD22, CD79A, CR2, FCRL2, IGKC, MS4A1 and PAX5	Transcriptional data from TCGA and GEO	Not reported	Positively correlated with overall survival and response to pembrolizumab
Italiano et al. [32]	Mixed STS	CD20 and CD3 to identify TLS	IHC	30/249 in prospective group 1/41 in all-comers retrospective group	Positively correlated with six-month, non-progression rate and progression-free survival

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
