# Peer review of "Cancer-Associated B Cells in Sarcoma"

_cancers, 2023, doi:10.3390/cancers15030622_

Round 1
Reviewer 1 Report
This is an excellent, well written and focused review on B cells in sarcoma, covering their detection, pronostic impact and predictive impact for response to immunotherapy, spatial organization in the tumor microenvironment and functions. It also nicely and adequately refers to "old" litterature, which is extremely uncommon but appreciable.
I have noticed
-1 error in ref 17 line 96 : this reference is on PDAC : it should be on sarcoma
-1 error line 128 : .."are thought to be mediated by follicular dendritic cell tumor antigen presentation to T cells ": follicular should be removed : this refers to antigen presentation to T cells by DCs ;
-line 93 : Mature TLS can be found in UPS (Petitprez et al, ext data fig 8)
Author Response
Thank you for your review and helpful comments.
We have made edits to address the errors and suggestions listed, including:
- Removing reference 17 from previous line 96
- Removing "follicular" from that sentence
- Adding a comment about mature TLS in UPS in Petitprez et al extended figure 8
Thank you
Reviewer 2 Report
Overall a pretty good review of literature and historical perspective. A few areas of deficiency need to be addressed. These include a better description of TLS organization and its relation to function evolution and selection of B cells in the TME, enlarging and more descriptive figures and more attention to the description of intratumoral Igs.
Figures
Figure 1. The figures and text need to be enlarged and is too small. The figure needs to be expanded and much more detail needs to be included. While the biorender images are a nice start, better description of the relevant structures in each image, the events and mechanism that drives each event in the functions of the B cells has to be included. For example which isotypes are driving what events, relevant Fc receptors, how antigen would be processed internalized and present or cross presented.
Figure 2 is also very rudimentary. A key to TLS function is the organization of the structure. In the review there’s references to mature and immature and its known this coincides with structural organization of the TLS. Both the figure and the legend needs to describe the key features and functions that the organization of cells drives and the phenotype (for example affinity maturation and differentiation into plasma or effector memory with mature or lack of with immature)
TLS
Since TLS are referred to throughout the review and in depth in many studies a more detailed description (including the figure) of their organization (mature versus immature) and how it relates to B cell function is needed.
Intratumoral sarcoma immunoglobulin.
This section is not straightforward. The way written it is be a bit confusing, is the author implying tumor cell expressed Igs are functional and play a role in tumor biology? The authors also need a more rigorous analysis of what the literature actually demonstrates before implying that Igs “expressed by sarcomas” are functional, evolved and relevant to anti-tumor function as those from highly regulated B cell development processes
There’s been quite a bit of analysis and literature on isotype and connection to anti-tumor function in many different types of cancer. This section needs better development. The description here of whether there is a connection of isotypes and response in sarcomas is not clear.
Author Response
Thank you very much for your insightful review and comments to improve our review paper.
We have addressed the comments below in the following manner:
- Figure 1 has been revamped and edited to include more detail include more information regarding specific receptor/ligand relationships, immunoglobulin subtypes and antigen cross-presentation.
- Figure 2 likewise has been edited to include much more detail. Here, we have provided more information about the micro-architecture of the mature TLS, including demonstration of effector functions in both T-cell and B-cell zones (including class-switching and differentiation of B-cells
- Accordingly, we have expanded our section on tertiary lymphoid structures. (track changed in edited version) - including more discussion regarding TLS maturity.
- Intratumoral immunoglobulin. We agree with your comments. We have greatly expanded this section. This includes an expanded discussion of the importance of Ig isotype and anti-tumor function, and a further discussion and analysis of the research outlining intrinsic tumor production of Ig.
Thank you